Rider weed deep residual network-based incremental model for text classification using multidimensional features and MapReduce

Abdalla Hemn Barzan hemn.db85@yahoo.com habdalla@kean.edu 1
Ahmed Awder M. 2
Zeebaree Subhi R.M. 3
Alkhayyat Ahmed 4
Ihnaini Baha 5
1 Department of Computer Science, Wenzhou-Kean University , Wenzhou , Zhejiang , China
2 Department of Communication Engineering, Technical College of Engineering, Sulaimani Polytechnic University , Sulaymaniyah , Iraq
3 Energy Department, Technical Collage Engineering, Duhok Polytechnic University , Duhok , Iraq
4 Department of Computer Technical Engineering, College of Technical Engineering, Islamic University , Najaf , Iraq
5 Department of Computer Science, Wenzhou-Kean University , Wenzhou , Zhejiang , China
Kong Xiangjie
Electronic publication date: 2022 Mar 31
Publication date: 2022
Volume: 8
Electronic Location ID: e937
Received 2021 Oct 21; Accepted 2022 Mar 8
Copyright: ©2022 Abdalla et al.
Copyright year: 2022
Copyright holder: Abdalla et al.
License: This is an open access article distributed under the terms of the Creative Commons Attribution License, which permits unrestricted use, distribution, reproduction and adaptation in any medium and for any purpose provided that it is properly attributed. For attribution, the original author(s), title, publication source (PeerJ Computer Science) and either DOI or URL of the article must be cited.
License URL: https://creativecommons.org/licenses/by/4.0/

Keywords: Text classification, Deep Residual network, MapReduce Model, Dynamic learning, Fuzzy theory

Funding: Leading Talents of Provincial Colleges and Universities, Zhejiang-China #WB20200915000043 Leading Talents of Provincial Colleges and Universities, Zhejiang-China (#WB2020091500 0043), supported this work. The funders had no role in study design, data collection and analysis, decision to publish, or preparation of the manuscript.

==============================
Increasing demands for information and the rapid growth of big data have dramatically increased the amount of textual data. In order to obtain useful text information, the classification of texts is considered an imperative task. Accordingly, this article will describe the development of a hybrid optimization algorithm for classifying text. Here, pre-processing was done using the stemming process and stop word removal. Additionally, we performed the extraction of imperative features and the selection of optimal features using the Tanimoto similarity, which estimates the similarity between features and selects the relevant features with higher feature selection accuracy. Following that, a deep residual network trained by the Adam algorithm was utilized for dynamic text classification. Dynamic learning was performed using the proposed Rider invasive weed optimization (RIWO)-based deep residual network along with fuzzy theory. The proposed RIWO algorithm combines invasive weed optimization (IWO) and the Rider optimization algorithm (ROA). These processes are carried out under the MapReduce framework. Our analysis revealed that the proposed RIWO-based deep residual network outperformed other techniques with the highest true positive rate (TPR) of 85%, true negative rate (TNR) of 94%, and accuracy of 88.7%.

Introduction

The massive demand for big data has necessitated the evaluation of their sources and implications. The fundamental opinion of analysis relies on designing a novel framework for studying the data. The similarity measure is one of the mathematical models used for classifying and clustering data. Here, we provide the fundamental assessment of common similarity measures such as Jaccard (Gonzalez, Bonventi Jr & Vieira Rodrigues, 2008), cosine (Tata & Patel, 2007), Euclidean distance (Schoenharl & Madey, 2008), and extended Jaccard (Kingma & Ba, 2014), which are utilized for evaluating distance or angle across vectors. In this article, the similarity measures are divided into feature content or topological metrics.

In topology, the features are organized in a hierarchical model and the appropriate path length across the features must be evaluated. The features are measured based on evidence: features with elevated frequency have explicitly elevated information, while features with lower frequency are adapted with less information. Pair-wise and ITSim metrics fit into the class of feature content metrics. The information content measure provides elevated priority to the highest features with a small difference between the two data, leading to improved outcomes. The cosine and Euclidean belong to the class of topological metrics. It is susceptible to information loss as two similar datasets can be significantly offset by the existence of solitary features (Kuppili et al., 2018). Methods such as clustering and classification that are utilized in text mining-based applications can also help transform massive data into small subsets to increase computational effectiveness (Kotte, Rajavelu & Rajsingh, 2019).

Text data consist of noisy and irrelevant features that prevent learning techniques from improving their accuracy. To remove redundant data, various data mining methods have been adapted. Feature extraction and selection are two such methods used to classify data. The selection of components is utilized for eliminating extra text features for effective classification and clustering. The previous techniques transformed huge data into small data while taking classical distance measures into consideration. Reducing dimensionality minimizes evaluation time and maximizes the efficiency of classification. The recovery of data and text are utilized in detecting data synonyms and meaning. Several techniques have been devised for the classification and clustering processes. Clustering is carried out using unsupervised techniques with different class label data (Kotte, Rajavelu & Rajsingh, 2019). The goal of classifying text is to categorize data into different parts. In this study, the goal was to allocate pertinent labels based on content (Wang et al., 2019).

Categorizing texts is considered a crucial part of processing natural language. It is extensively employed in applications such as automatic medical text classification (Ali et al., 2021b) and traffic monitoring (Ali et al., 2021a). Most news services require repeated arrangement of numerous articles in a single day (Lai et al., 2015). Advanced email services offer the function of sorting junk mail and mail in an automated manner (Wu et al., 2017). Other applications involve the analysis of sentiment (Mäntylä, Graziotin & Kuutila, 2018), modeling topics (He et al., 2017), text clustering  (Vidyadhari, Sandhya & Premchand, 2019), translation of languages (Wu et al., 2016), and intent detection (Kim et al., 2016; Wang et al., 2019). Technology classification assists people, filters useful data, and poses more implications in real life. The design of text categorization and machine learning (Ali et al., 2017; Ali, El-Sappagh & Kwak, 2019) has shifted from manual to machine  (Chen, Yan & Wong, 2018; Cheng & Malhi, 2017; Esteva et al., 2017; Luo, 2017). Several textualization classification techniques exist  (Zhang, Li & Du, 2017) with the goal of categorizing textual data. The categorization outcomes can fulfill an individual’s requirements for classifying text and are suitable for rapidly attaining significant data. MapReduce is utilized for handling huge amounts of data using unstructured data  (Liu & Wang, 2020).

Our aim is to devise an optimization-driven deep learning technique for classifying texts using the MapReduce framework. First, the text data underwent pre-processing to remove unnecessary words. Pre-processing was performed using stop word removal and the stemming process. After that, we extracted the features, such as SentiWordNet, thematic, and contextual features. These features were employed in a deep residual network for classifying the texts and the deep residual network training was performed using the Adam algorithm. Finally, dynamic learning was carried out wherein the proposed Rider invasive weed optimization (RIWO)-based deep residual network was employed for incremental text classification. The fuzzy theory was employed for weight bounding to deal with the incremental data. In this process, the deep residual network training was performed using the proposed RIWO, which was devised by combining the Rider optimization algorithm (ROA) and invasive weed optimization (IWO) algorithm.

The key contributions of this paper are:

• Proposed RIWO-based deep residual network for text classification: A new method developed using multidimensional features and MapReduce. Dynamic learning uses the proposed RIWO-based deep residual network for classifying texts. Here, the developed RIWO was adapted for deep residual network training.

• RIWO: Devised by combining ROA and IWO algorithms.

• The fuzzy theory: Employed to handle dynamic data by performing weight bounding.

The rest of the sections are given as follows: ‘Literature Review’ presents the classical text classification techniques survey. ‘Proposed RIWO-based Deep Residual Network for Text Classification in Big Data’ describes the developed text classification model. ‘Results and Discussion’ discusses the results of the developed model for classical techniques, and ‘Conclusion’ presents the conclusion.

Literature review

The eight classical techniques based on text classification using big data and their issues are described below. Ranjan & Prasad (2018) developed an LFNN-based incremental learning technique for classifying text data based on context-semantic features. The methods employed a dynamic dataset for classification to dynamically learn the model. Here, we employed the incremental learning procedure Back Propagation Lion (BPLion) Neural Network, and fuzzy bounding and the Lion algorithm (LA) were used to select the weights. However, the technique failed to precisely classify the sentence. Kotte, Rajavelu & Rajsingh (2019) devised a similarity function for clustering the feature pattern. The technique attained dimensionality reduction with improved accuracy. However, the technique failed to utilize membership functions for obtaining clusters. Wang et al. (2019) devised a deep learning technique for classifying text documents. Additionally, a large-scale scope-based convolutional neural network (LSS-CNN) was utilized for categorizing the text. The method effectively computed scope-based data and parallel training for massive datasets. The technique attained high scalability on big data but failed to attain the utmost accuracy. Kuppili et al. (2018) developed the Maxwell–Boltzmann Similarity Measure (MBSM) for classifying text. The MBSM was derived with feature values from the documents. The MBSM was devised by combining single label K-nearest neighbor’s classification (SLKNN), multi-label KNN (MLKNN), and K-means clustering. However, the technique failed to include clustering techniques and query mining.  Liu & Wang (2020) devised a technique for classifying text using English quality-related text data. Here, the goal was to extract, classify, and examine the data from English texts while considering cyclic neural networks. Ultimately, the features with sophisticated English texts were generated. This technique also combined attention to improve label disorder and make the structure more reliable. However, the computation cost tended to be very high. Qi et al. (2020) designed a method for classifying text and solving the misfitting issue by performing angle-pruning tasks from a database. The technique computed the efficiency of each convolutional filter using discriminative power produced at the pooling layer and shortened words obtained from the filter. However, the technique produced high computational complexity.

 BenSaid & Alimi (2020) devised the Multi-Objective Automated Negotiation-based Online Feature Selection (MOANOFS) for classifying texts. The MOANOFS utilized automated negotiation and machine learning techniques to improve classification performance using ultra-high dimensional datasets. This helped the method to decide which features were the most pertinent. However, the method failed to select features from multi-classification domains. Jiang et al. (2018) devised a hybrid text classification model based on softmax regression for classifying text. The deep belief network was utilized to classify text using learned feature space. However, the technique failed to filter extraneous characters for enhancing system performance.

 Akhter et al. (2020) developed a large multi-format and multi-purpose dataset with more than ten thousand documents organized into six classes. For text classification, they utilized a Single-layer Multisize Filters Convolutional Neural Network (SMFCNN). The SMFCNN obtained high accuracy, demonstrating its capability to classify long text documents in Urdu. Flores, Figueroa & Pezoa (2021) developed a query strategy and stopping criterion that transformed Classifier Regular Expression (CREGEX) in an active learning (AL) biomedical text classifier. As a result, the AL was permitted to decrease the number of training examples required for a similar performance in every dataset compared to passive learning (PL).

 Huan et al. (2020) introduced a method for Chinese text classification that depended on a feature-enhanced nonequilibrium bidirectional long short-term memory (Bi-LSTM) network. This method enhanced the precision of Chinese text classification and had a reliable capability to recognize Chinese text features. However, the accuracy of Chinese text recognition needs improvement and the training processing time should be reduced. Dong et al. (2019) introduced a text classification approach using a self-interaction attention mechanism and label embedding. This method showed high classification accuracy, but for practical application, more work should be done.

Proposed RIWO-based deep residual network for text classification in big data

The objective of text classification is to categorize text data into different classes based on certain content. Text classification is considered an imperative part of processing natural language. However, it is considered a challenging and complex process due to high dimensional and noisy texts, and the need to devise an improved classifier for huge textual data. This study devised a novel hybrid optimization-driven deep learning technique for text classification using big data. Here, the goal was to devise a classifier that employs text data as input and allocates pertinent labels based on content. At first, the input text data underwent pre-processing to eliminate noise and artifacts. Pre-processing was performed with stop word removal and stemming. Once the pre-processed data were obtained, the contextual, thematic, and SentiWordNet features were extracted. Once the features were extracted, the imperative features were chosen using the Tanimoto similarity. The Tanimoto similarity method evaluates similarity across features and chooses the relevant features with high feature selection accuracy. Once the features were selected, a deep residual network  (Chen et al., 2019) was used for dynamic text classification. The deep residual network was trained using the Adam algorithm (Kingma & Ba, 2014; Abdalla, Ahmed & Al Sibahee, 2020; Mohsin, Li & Abdalla, 2020). Additionally, dynamic learning was performed using the proposed RIWO algorithm along with the fuzzy theory. The proposed RIWO algorithm integrates IWO (Sang, Duan & Li, 2018) and ROA (Binu & Kariyappa, 2018). Figure 1 shows the schematic view of text classification from the input text data in big data using the proposed RIWO method, considering the MapReduce phase.

Figure 1 Schematic view of text classification from the input big data using proposed RIWO-based Deep Residual Network.

Assume input text data with various attributes is expressed as: (1) B=Bd,e;1≤d≤D1≤e≤E

where, Bd,e refers to text data contained in the database with an attribute in data. Data points are employed using attributes for each data point. The other step was to eliminate artifacts and noise present in the data.

The data in a database are split into a specific number that is equivalent to mappers present in the MapReduce model. The partitioned data is given by: (2) Bd,e=Dq;1≤q≤N

where, N symbolizes total mappers. Assume mappers in MapReduce are expressed as: (3) M=M1,M2,…,Mq,…,MN;1≤q≤N

Thus, input to mapper is formulated as: (4) Dq=dr,l;1≤r≤mq;1≤l≤n

where dr,l symbolizes split data given to mapper, and Dq indicates data in mapper.

Pre-processing

The partitioned data from the text dataset was pre-processed by removing stop words and using stemming. Pre-processing is an important process used to smoothly arrange various data and offer effective outcomes by improving representation. The dataset contains unnecessary phrases and words that influence the process. Therefore, pre-processing is important for removing inconsistent words from the dataset. Initially, the text data are accumulated in the relational dataset and all reviews are divided into sentences and bags of the sentence. The elimination of stop words is carried out to maximize the performance of the text classification model. Here, the stemming and stop word removal refined the data.

Stop word removal

This is a process that removes words with less representative value for the data. Some of the non-representative words include pronouns and articles. When evaluating data, some words are not valuable to text content, and removing such redundant words is imperative. This procedure is termed stop word removal (Dave & Jaswal, 2015). Certain words such as articles, conjunctions, and prepositions, continuously appear and are called stop words. The removal of the stop word, the most imperative technique, is utilized to remove redundant words using vocabulary because the vector space size does not offer any meaning. The stop words indicate the word, which does not hold any data. It is a process used to eliminate stop words from a large set of reviews. The elimination of the stop word is used to save space and accelerate and improve processing.

Stemming

The stemming procedure is utilized to convert words to stem form. In massive amounts of data, several words are utilized that convey a similar meaning. Therefore, the critical method used to minimize words to root is stemming. Stemming is a method of linguistic normalization wherein little words are reduced. Moreover, it is the procedure used to retrieve information for describing the mechanism of reducing redundant words to their root form and word stem. For instance, the words connections, connection, connecting, and connected are all reduced to connect (Dave & Jaswal, 2015). (5) Ml=Qi,1≤η≤Pk

where Qi symbolizes total words present in text data from the database.

The pre-processed outcome generated from pre-processing is expressed as Ml, which is subjected as an input to feature extraction phase.

Acquisition of features for producing highly pertinent features

This describes an imperative feature produced with input review, and the implication of feature extraction is used to produce pertinent features that facilitate improved text classification. Moreover, data obstruction is reduced because text data is expressed as a minimized feature set. Therefore, the pre-processed partitioned data is fed to feature extraction, wherein SentiWordNet, contextual, and thematic features are extracted.

Extraction of SentiWordNet features

The SentiWordNet features are utilized from pre-processed partitioned data by removing keywords from reviews. Here, the SentiWordNet (Ghosh & Kar, 2013) is employed as a lexical resource to extract the SentiWordNet features. The SentiWordNet assigns each WordNet text one of three numerical sentiment scores: positive, negative, or neutral. Here, different words indicated different polarities that indicated various word senses. The SentiWordNet consisted of different linguistic features: verbs, adverbs, adjectives, and n-gram features. SentiWordNet is a process used to evaluate the score of a specific word using text data. Here, the SentiWordNet was employed to determine the polarity of the offered review and for discovering positivity and negativity. Hence, SentiWordNet is modeled as F1.

Extraction of contextual features

The context-based features (Ranjan & Prasad, 2018) were generated from pre-processed partitioned data that described relevant words by dividing them using non-relevant reviews for effective classification. This requires finding key terms that have context and semantic meaning in order to establish a proper context. The key term is considered a preliminary indicator for relevant review while context terms act as a validator that can be used to evaluate if the key term is an indicator. Here, the training dataset contained keywords with pertinent words. The context-based features assisted in selecting the relevant and non-relevant reviews.

We considered representing a training dataset that had relevant and non-relevant reviews. Using this method, assume xs represents the key term and xc indicates the context term.

- Detection of key terms:

The language model that employs each term and the metric are expressed as: (6) C=LrelLon_rel

where, Lrel symbolizes the language model for Nrel and Lnon-rel signifies the language model for Nnon_rel.

- Discovery of the context term:

After discovering key terms, the process of context term discovery, which is similar to separately detecting each term, begins. The steps employed in determining the context term are given as:

(i) Computing all instances of the key term employed among relevant and non-relevant reviews.

(ii) By employing sliding window size, the conditions are mined as context terms. Hence, the size of the window is employed as a context span.

(iii) The pertinent terms generated are employed as a text, modeled as dr, and non-relevant terms are denoted as dnr. The set of pertinent text is modeled as Rd_r, and the non-relevant set is referred to as Rd_nr.

(iv) After that, the score evaluated for each distinctive term is expressed as: (7) CxC=|LRd_rxC−LRd_nrxC|S

where LRd_r(xC) symbolizes the language model for an excerpt with relevant review set. The term LRd_nr(xC) indicates a language model for an excerpt with a non-relevant review set and S represents the size of the window. If the measure is a definite threshold then that score is adapted as a context term xS. Generated context-based features are modeled as F2.

Extraction of thematic features

The pre-processed partitioned data dr,l isused to find thematic features. Here, the count of the thematic word (Tas & Kiyani, 2007) in a sentence is imperative as the frequently occurring words are most likely connected to the topic in the data. Thematic words are words that grab key topics defined in a provided document. In thematic features, the top 10 most frequent words are employed as thematic. Thus, the thematic feature F3 is modeled as: (8) F1=T∑t=0ℓTt

where T expresses the count of thematic words in a sentence, and it is expressed as T1,T2,.…,Tℓ.

The feature vectors considering the contextual, thematic, and SentiWordNet features are expressed as: (9) F=F1,F2,F3

where F1 symbolizes SentiWordNet features, F2 signifies contextual features, and F3 refers to thematic features.

Feature selection using the Tanimoto similarity

The selection of imperative features from the extracted features F is made using the Tanimoto similarity. The Tanimoto similarity computes similarity across features and selects features with high feature selection accuracy. Here, the Tanimoto similarity is expressed as: (10) S=∑w=1mywzw ∑w=1myw2+ ∑w=1mzw2− ∑w=1mywzw

where S indicates the Tanimoto measure yw and zw represents features. The selected features are expressed as R.

The produced feature selection output obtained from the mapper is input to the reducer U. Then, the text classification is performed on the reducer using the selected features, which is briefly illustrated below.

Classification of texts with Adam-based deep residual network

Text classification is performed using an Adam-based deep residual network and selected features R. The classification of text data assists in standardizing the infrastructure and makes the search simpler and more pertinent. Additionally, classification enhances the user’s experience, simplifies navigation, and helps solve business issues (such as social media and e-mails) in real-time. The deep residual network is more effective at counting attributes and computation. This network is capable of building deep representations at each layer and can manage advanced deep learning tasks. The architecture of the deep residual network and training with the Adam algorithm is described below.

Architecture of the deep residual network

We employed a deep residual network (Chen et al., 2019) in order to make a productive decision regarding which text classification to perform. The DRN is comprised of different layers: residual blocks, convolutional (Conv) layers, linear classifier, and average pooling layers. Figure 2 shows the structural design of a deep residual network with residual blocks, Conv layers, linear classifier, and average pooling layers for text classification.

Figure 2 Structural design of deep residual network with residual blocks, convolutional (Conv) layers, linear classifier, and average pooling layers for text classification.

-Conv layer:

The two-dimensional Conv layer reduces free attributes in training and offers reimbursement for allocating weight. The cover layer processes the input image with the filter sequence known as the kernel using a local connection. The cover layer utilizes a mathematical process for sliding the filter with the input matrix and computes the dot product of the kernel. The evaluation process of the Conv layer is represented as: (11) B2dO= ∑a=0E−1 ∑s=0E−1Xa,s⋅Ou+a,v+s

(12) B1dO= ∑Z=0Cin−1GZ∗O

where O expresses the CNN feature of the input image, u and v refer to the recording coordinates, G signifies the E × E kernel matrix termed as a learnable parameter, and a and s are the position indices of the kernel matrix. Hence, GZ expresses the size of the kernel for the Zth input neuron and ∗ expresses the cross-correlation operator.

Pooling layer: This layer is associated with the Conv layer and is especially utilized to reduce the feature map’s spatial size. The average pooling is selected as a function of each slice and the depth of the feature map. (13) aout=ain−Zaλ+1

(14) sout=sin−Zsλ+1

where ain symbolizes the input matrix width, sin signifies the height of the input matrix, aout and sout represent the respective value of output, and Za and Zs symbolize the width and height of the kernel size.

-Activation function: The nonlinear activation function is adapted for learning nonlinear and complicated features so it is utilized to improve the non-linearity of extracted features. Rectified linear unit (ReLU) is utilized for processing data. The ReLU function is formulated as: (15) ReLUO=0;K<0K;K≥0

where K symbolizes a feature.

-Batch normalization: Here, the training set was divided into various small sets known as mini-batches to train the model. It attains a balance between evaluation and convergence complexity. The input layers are normalized by scaling activations to maximize reliability and training speed.

-Residual blocks: This indicates the shortcut connection amongst the Conv layers. The input is unswervingly allocated to output only if input and output are of equal size. (16) κ=ℜO+O

(17) κ=ℜO+ƛMO

where O and κ signify input and output residual blocks, κ symbolizes mapping relation, ƛM expresses dimension matching factor, and ℜ. signifies activation function.

-Linear classifier: After completion of the Conv layer, linear classifier performs a procedure to discover noisy pixels using input features. It is a combination of the softmax function and a fully connected layer. (18) κ=ƛκ+υ

where ƛ expresses weight matrix and υ represents bias. Figure 2 shows the structural design of the deep residual network. Here, the output, represented as κ, assists in classifying the texts.

Training of the deep residual network with the Adam algorithm

The deep residual network training is performed using the Adam technique which assists in discovering the best weights for tuning the deep residual network for classifying text. Adam (Kingma & Ba, 2014) represents a first-order stochastic gradient-based optimization extensively adapted to a fitness function that changes for attributes. The major implication of the method is computational efficiency and fewer memory needs. Moreover, the problems associated with the non-stationary objectives and the subsistence of noisy gradients are handled effectively. In this study, the magnitudes of the updated parameters were invariant in contrast to the rescaling of gradient, and step size was handled with a hyperparameter that worked with sparse gradients. In addition, Adam is effective in performing step size annealing. The classification of text employs a deep residual network for texts. The steps of Adam are given as:

Step 1: Initialization

The first step represents bias correction initialization wherein q ˆl signifies the corrected bias of the first moment estimate and m ˆl represents the corrected bias of the second moment estimate.

Step 2: Discovery of error

The bias error is computed to choose the optimum weight for training the deep residual network. Here, the error was termed as an error function that led to an optimal global solution. The function is termed as a minimization function and is expressed as: (19) Err=1f∑l=1fOl−κ2

where f signifies total data, κ symbolizes output generated with the deep residual network classifier, and Ol indicates the expected value.

Step 3: Discovery of updated bias

Adam is used to improving convergence behavior and optimization. This technique generates smooth variation with effectual computational efficiency and lower memory requirements. As per Adam (Kingma & Ba, 2014), the bias is expressed as: (20) θl=θl−1−αq ˆlm ˆl+ɛ

where α refers to step size, q ˆl expresses corrected bias, m ˆl indicates bias-corrected second-moment estimate, ɛ represents the constant, and θl−1 signifies the parameter at a prior time instant (l − 1). The corrected bias of the first-order moment is expressed as: (21) q ˆl=ql1−η1l

(22) q ˆl=η1ql−1+1−η1Gl1.

The corrected bias of the second-order moment is represented as: (23) m ˆl=ml1−η2l

(24) m ˆl=η2ml−1+1−η2Hl2

(25) Hl=∇θlossθl−1.

Step 4: Determination of the best solution: The best solution is determined with error, and a better solution is employed for classifying text.

Step 5: Termination: The optimum weights are produced repeatedly until the utmost iterations are attained. Table 1 describes the pseudocode of the Adam technique.

Table 1 Pseudocode of the Adam algorithm.

Input: q ˆl: corrected bias of first moment estimate and m ˆl: corrected bias of second moment estimate	
Output: Resulting parameters	
Require α step size	
Require η1, η2 ∈ [0, 1]: Exponential rate decay	
Initialize first-moment vector	
Initialize second-moment vector	
Initialize time step	
While θl not converged do	
l←l + 1	
Get gradients	
Update first biased moment estimate using Eq. (22)	
Update second biased moment estimate using Eq. (24)	
Evaluate corrected bias first-moment estimate	
Evaluate corrected bias second-moment estimate	
Update parameters	
End while	
Return θl	

Dynamic learning with the proposed RIWO-based deep residual network

For incremental data B, dynamic learning is done using the proposed RIWO-based deep residual network. Here, the assessment of incremental learning with the developed RIWO-based deep residual network was done to achieve effective text classification with the dynamic data. The deep residual network was trained with developed RIWO for generating optimum weights. The developed RIWO was generated by integrating ROA and IWO for acquiring effective dynamic text classification.

Architecture of deep residual network

The model of the deep residual network is described in ‘Classification of Texts with Adam-Based Deep Residual Network’.

Training of deep residual network with proposed RIWO

The training of the deep residual networks was performed with the developed RIWO, which was devised by integrating IWO and ROA. Here, the ROA (Binu & Kariyappa, 2018) was motivated by the behavior of the rider groups, which travel and compete to attain a common target position. In this model, the riders were chosen from the total number of riders for each group. We concluded that this method produces enhanced classification accuracy. Furthermore, the ROA is effective and follows the steps of fictional computing for addressing optimization problems but with less convergence. IWO (Sang, Duan & Li, 2018) is motivated by colonizing characteristics of weed plants. The technique showed a fast convergence rate and elevated the accuracy. Hence, we integrated IWO and ROA to enhance complete algorithmic performance. The steps in the method are expressed as:

Step (1) Initialization of population

The preliminary step is algorithm initialization, which is performed using four-rider groups provided by A and represented as: (26) A=A1,A2,…,Aμ,…,Aϑ

where Aμ signifies μth rider and ϑ isthe total riders.

Step (2) Determination of error:

The computation of errors is already described in Eq. (19).

Step (3) Update the riders’ position:

The rider position in each set is updated for determining the leader. Thus, the updated rider position using a feature of each rider is defined below. The updated position of each rider is expressed as:

As per ROA (Binu & Kariyappa, 2018), the updated overtaker position is used to increase the rate of success by determining the position of the overtaker and is represented as: (27) An+1og,h=Anog,h+∂n∗g∗ALL,h

where ∂n∗g signifies the direction indicator.

The attacker has a propensity to grab the position of the leaders and is given by: (28) An+1ag,u=ALL,u+CosKg,un∗ALL,u+rgn.

The bypass riders contain a familiar path, and its update is expressed as: (29) An+1bg,u=λAnχ,u∗δu+Anξ,u∗1−δu

where λ symbolizes the random number, χ signifies the arbitrary number between 1 and P, ξ denotes an arbitrary number between 1 and P, and δ expresses an arbitrary number between 0 and 1.

The follower poses a propensity to update the position using the leading rider position to attain the target and is given by: (30) An+1Fg,h=ALL,h+CosKg,hn∗ALL,h∗rgn

where h is the coordinate selector, AL indicates the leading rider position, L represents the leading rider index, Kg,hn represents the steering angle of gth rider in hth coordinate, and rgn is the distance. (31) An+1Fg,h=ALL,h1+CosKg,hn∗rgn.

The IWO assists in generating the best solutions. Per IWO (Sang, Duan & Li, 2018), the equation is represented as: (32) An+1Fg,h=σnAnF+A best−AnF

where An+1F symbolizes the new weed position in iteration n + 1, AnF signifies the current weed position, Abest refers to the best weed found in the whole population, and σ(n) represents the current standard deviation. (33) Abest=An+1Fg,h−σnAnF+AnF.

Substitute Eq. (33) in Eq. (31), (34) An+1Fg,h=An+1Fg,h−σnAnF+AnF1+CosKg,hn∗rgn

(35) An+1Fg,h=An+1Fg,h1+CosKg,hn∗rgn−σnAnF+AnF1+CosKg,hn∗rgn

(36) An+1Fg,h−An+1Fg,h1+CosKg,hn∗rgn=AnF−σnAnF1+CosKg,hn∗rgn

(37) An+1Fg,h1−1−CosKg,hn∗rgn=AnF−σnAnF1+CosKg,hn∗rgn

(38) An+1Fg,h−CosKg,hn∗rgn=AnF−σnAnF1+CosKg,hn∗rgn.

The final updated equation of the proposed RIWO is expressed as: (39) An+1Fg,h=−AnF−AnFσn1+CosKg,hn∗rgnCosKg,hn∗rgn.

Step (4) Re-evaluation of the error:

After completing the update process, the error of each rider is computed. The position of the rider in the leading position is replaced using the position of the new generated rider so that the error of the new rider is smaller.

Step (5) Update of the rider parameter:

The rider attribute update is imperative to determine an effectual optimal solution using the error.

Step (6) Riding off time:

The steps were iterated repeatedly until we attained off time NOFF, in which the leader was determined. The pseudocode of the developed RIWO is shown in Table 2.

Table 2 Pseudocode of developed RIWO.

Input: A: Arbitrary rider position, n: iteration, nmax: maximum iteration	
Output: Leader AL	
Begin	
Initialize solutions set	
Initialize algorithmic parameter	
Discover error using Eq. (19)	
While n < NOFF	
For ν = 1 to P	
Update bypass position with Eq. (29)	
Update follower position with Eq. (39)	
Update overtaker position with Eq. (27)	
Update attacker position with Eq. (28)	
Rank riders using error with Eq. (19)	
Choose the rider with minimal error	
Update steering angle, gear, accelerator, and brake	
Return AL	
n = n + 1	
End for	
End while	
End	

The output produced from the developed RIWO-based deep residual network is κ, which helps classify the text data since dynamic learning helps classify the dynamic data. Here, we employed fuzzy bounding to remodel the classifier if there was a high chance of a previous data error.

Fuzzy theory

An error is evaluated whenever incremental data is added to the model and weights are updated without using the previous weights. If the error evaluated by the present instance is less than the error of the previous instance then the weights are updated based on the proposed RIWO algorithm. Otherwise, the classifiers are remodeled by setting a boundary for weight using fuzzy theory (Ranjan & Prasad, 2018) and optimal weight is chosen using the proposed RIWO algorithm. On arrival of data di+1, the error ei+1 will be computed and compared with that of the previous data di. If ei < ei+1, then prediction with training based on RIWO is made. Otherwise, the fuzzy bounding-based learning will be done by bounding the weights, which is given as: (40) ωB=ωt±Fs

where ωt is weight at the current iteration and Fs signifies a fuzzy score. For the dynamic data, the features {F} were extracted. Here, the membership degree is given as: (41) Membership Degree=||ωt−2−ωt−1||

where ωt−2 represents weights at iteration t − 2 and ωt−1 signifies weights at iteration t − 1. When the highest iteration is attained, the process is stopped.

Results and Discussion

The competence of the technique is evaluated by analyzing the techniques using various measures like true positive rate (TPR), true negative rate (TNR), and accuracy. The assessment is done by considering mappers = 3, mappers = 4, and by varying the chunk size.

Experimental setup

The execution of the developed model was performed in PYTHON with Windows 10 OS, an Intel processor, and 4GB RAM. Here, the analysis was performed by considering the NSL-KDD dataset.

Dataset description

The dataset adapted for text classification involved the Reuters and 20 Newsgroups databases and is explained below.

The 20 Newsgroups database

The 20 Newsgroups dataset (Crawford, 2020) was curated by Ken Lang for newsreaders to extract the Netnews. The dataset was established by collecting 20,000 newsgroup data points split across 20 different newsgroups. The database is popular for analyzing text applications used to handle machine-learning methods such as clustering and text classification. The dataset is organized into 20 different newsgroups covering different topics.

Reuters database

The Reuters-21578 Text Categorization Collection Dataset was curated by David D. Lewis (NLTK Data, 2020). The dataset is comprised of documents collected from Reuters newswires starting in 1987. The documents are arranged and indexed based on categories. There were 21,578 instances in the dataset with five attributes. The number of websites attained by the dataset was 163,417.

Evaluation metrics

The efficiency of the developed model was examined by adopting measures such as accuracy, TPR, and TNR.

Accuracy

Accuracy is described as the measure of data that is precisely preserved and is expressed as: (42) Acc=P+QP+Q+H+F

where P signifies true positive, Q symbolizes true negative, H denotes true false positive, and F is a false negative.

TPR

The TPR refers to the ratio of the count of true positives with respect to the total number of positives. (43) TPR=PP+H

where P refers to true positives and H is the false negatives.

TNR

The TNR refers to the ratio of negatives that are correctly detected. (44) TNR=QQ+F

where Q is true negative and F signifies false positive.

Comparative methods

We evaluated the proposed RIWO-based deep residual network by comparing it with other classical techniques such as LSS-CNN (Wang et al., 2019), RNN (Liu & Wang, 2020), SLKNN+MLKNN (Kuppili et al., 2018), BPLion+LFNN (Ranjan & Prasad, 2018), SVM, NN, and LSTM.

Comparative analysis

The proposed technique was assessed using certain measures such as accuracy, TPR, and TNR. Here, the analysis was performed by considering the Reuters and 20 Newsgroups datasets, as well as the mapper size = 3 and 4.

Analysis with the Reuters dataset

The assessment of techniques with the Reuters dataset considering TPR, TNR, and accuracy parameters is described. The assessment is done with mapper = 3 and mapper = 4 and varying the chunk size.

(a) Assessment with mapper = 3

Figure 3 shows an assessment of techniques measuring accuracy, TPR, and TNR in the Reuter datasets with mapper = 3. The assessment of techniques with the TPR measure is depicted in Fig. 3A. For chunk size = 3, the TPR evaluated by LSS-CNN, RNN, SLKNN+MLKNN, SVM, NN, LSTM, BPLion+LFNN, and the proposed RIWO-based deep residual network were 0.747, 0.757, 0.771, 0.776, 0.780, 0.785, 0.790, and 0.803, respectively. Likewise, for chunk size = 6, the TPR evaluated using LSS-CNN, RNN, SLKNN+MLKNN, SVM, NN, LSTM, BPLion+LFNN, BPLion+LFNN, and the proposed RIWO-based deep residual network were 0.800, 0.812, 0.818, 0.818, 0.818, 0.819, 0.819, and 0.830, respectively. The assessment of techniques measuring TNR is depicted in Fig. 3B. For chunk size = 3, the TNR evaluated using LSS-CNN, RNN, SLKNN+MLKNN, SVM, NN, LSTM, BPLion+LFNN, BPLion+LFNN, and the proposed RIWO-based deep residual network were 0.831, 0.842, 0.853, 0.861, 0.870, 0.880, 0.886, and 0.913, respectively. For chunk size = 6, the TNR evaluated using LSS-CNN, RNN, SLKNN+MLKNN, SVM, NN, LSTM, BPLion+LFNN, BPLion+LFNN, and the proposed RIWO-based deep residual network were 0.846, 0.850, 0.865, 0.869, 0.878, 0.885, 0.896, and 0.925, respectively. The assessment of the methods measuring accuracy is depicted in Fig. 3C. For chunk size = 3, the accuracy evaluated using LSS-CNN, RNN, SLKNN+MLKNN, SVM, NN, LSTM, BPLion+LFNN, BPLion+LFNN, and the proposed RIWO-based deep residual network were 0.8306, 0.842, 0.852, 0.861, 0.870, 0.880, 0.886, and 0.913, respectively. Likewise, for chunk size = 6, the accuracy evaluated using LSS-CNN, RNN, SLKNN+MLKNN, SVM, NN, LSTM, BPLion+LFNN, BPLion+LFNN, and the proposed RIWO-based deep residual network were 0.846, 0.850, 0.865, 0.869, 0.878, 0.885, 0.896, and 0.925, respectively. The performance improvement of LSS-CNN, RNN, SLKNN+MLKNN, SVM, NN, LSTM, BPLion+LFNN, BPLion+LFNN with respect to the proposed RIWO-based deep residual network considering accuracy were 8.540%, 8.108%, 6.486%, 6.054%, 5.081%, 4.324%, and 3.135%, respectively.

Figure 3 Assessment of different techniques comparing with the proposed method by considering Reuter dataset with mapper = 3. (A) TPR. (B) TNR. (C) Accuracy.

(b) Assessment with mapper = 4

We assessed the techniques by measuring accuracy, TPR, and TNR, considering the Reuters dataset and using mapper = 4 (Fig. 4). The assessment of techniques using TPR is displayed in Fig. 4A. For chunk size = 3, the TPR evaluated using LSS-CNN was 0.754, RNN was 0.768, SLKNN+MLKNN was 0.792, SVM was 0.796, NN was 0.800, LSTM was 0.806, BPLion+LFNN was 0.810, and the proposed RIWO-based deep residual network was 0.828. Likewise, for chunk size = 6, the TPR evaluated using LSS-CNN was 0.810, RNN was 0.820, SLKNN+MLKNN was 0.824, SVM was 0.824, NN was 0.825, LSTM was 0.826, BPLion+LFNN was 0.826, and the proposed RIWO-based deep residual network was 0.850. The assessment of techniques measuring TNR is depicted in Fig. 4B. For chunk size = 3, the TNR evaluated using LSS-CNN was 0.839, RNN was 0.860, SLKNN+MLKNN was 0.863, SVM was 0.870, NN was 0.875, LSTM was 0.881, BPLion+LFNN was 0.896, and the proposed RIWO-based deep residual network was 0.925. Likewise, for chunk size = 6, the TNR evaluated by LSS-CNN was 0.855, RNN was 0.856, SLKNN+MLKNN was 0.876, SVM was 0.878, NN was 0.885, LSTM was 0.893, BPLion+LFNN was 0.900, and the proposed RIWO-based deep residual network was 0.940. The assessment of the methods measuring accuracy is displayed in Fig. 4C. For chunk size = 3, the accuracy evaluated by LSS-CNN was 0.837, RNN was 0.843, SLKNN+MLKNN was 0.846, SVM was 0.850, NN was 0.855, LSTM was 0.858, BPLion+LFNN was 0.862, and the proposed RIWO-based deep residual network was 0.880. For chunk size = 6, the accuracy evaluated by LSS-CNN was 0.833, RNN was 0.849, SLKNN+MLKNN was 0.852, SVM was 0.857, NN was 0.859, LSTM was 0.863, BPLion+LFNN was 0.868, and the proposed RIWO-based deep residual network was 0.887. The performance improvement of LSS-CNN, RNN, SLKNN+MLKNN, SVM, NN, LSTM, and BPLion+LFNN with respect to the proposed RIWO-based deep residual network and considering accuracy was 6.087%, 4.284%, 3.945%, 3.38%, 3.156%, 2.705%, and 2.142%, respectively.

Figure 4 Assessment of different techniques comparing with the proposed method by considering Reuter dataset with mapper = 4. (A) TPR. (B) TNR. (C) Accuracy.

Analysis with the 20 Newsgroups dataset

The assessment of techniques using the 20 Newsgroups datasets with TPR, TNR, and accuracy parameters was elaborated. The assessment was done with mapper = 3 and mapper = 4 and by altering the chunk size.

(a) Assessment with mapper = 3

Figure 5 presents the assessment of techniques measuring accuracy, TPR and TNR and considering the 20 Newsgroups dataset with mapper = 3. The assessment of techniques measuring TPR is depicted in Fig. 5A. For chunk size = 3, the maximum TPR of 0.834 was determined using the proposed RIWO-based deep residual network, while the TPR found using LSS-CNN, RNN, SLKNN+MLKNN, SVM, NN, LSTM, and BPLion+LFNN were 0.708, 0.759, 0.780, 0.785, 0.792, 0.803, and 0.812, respectively. Likewise, for chunk size = 6, the highest TPR of 0.840 was found using the proposed RIWO-based deep residual network, while the TPR found using LSS-CNN, RNN, SLKNN+MLKNN, SVM, NN, LSTM, and BPLion+LFNN were 0.796, 0.815, 0.818, 0.822, 0.825, 0.828, and 0.829, respectively. The assessment of techniques measuring TNR is depicted in Fig. 5B. For chunk size = 3, the TNR computed using the proposed RIWO-based deep residual network was 0.862, while the TNR found using LSS-CNN, RNN, SLKNN+MLKNN, SVM, NN, LSTM, and BPLion+LFNN were 0.832, 0.839, 0.843, 0.846, 0.848, 0.850, and 0.851, respecitvely. Likewise, for chunk size = 6, the TNR evaluated using the proposed RIWO-based deep residual network was 0.879, while those computed by LSS-CNN, RNN, SLKNN+MLKNN, SVM, NN, LSTM, and BPLion+LFNN were 0.833, 0.840, 0.849, 0.808, 0.851, 0.852, and 0.854, respectively. The assessment of the methods measuring accuracy is depicted in Fig. 5C. For chunk size = 3, the accuracy evaluated by the proposed RIWO-based deep residual network was 0.850, while the accuracies computed by LSS-CNN, RNN, SLKNN+MLKNN, SVM, NN, LSTM, and BPLion + LFNN were 0.821, 0.822, 0.833, 0.837, 0.839, 0.840, and 0.843, respectively. The performance improvement of LSS-CNN, RNN, SLKNN+MLKNN, SVM, NN, LSTM, and BPLion + LFNN with respect to the proposed RIWO-based deep residual network considering accuracy were 3.411%, 3.294%, 2%, 1.529%, 1.294%, 1.17%, and 0.823%, respectively. Likewise, for chunk size = 6, the accuracy evaluated using the proposed RIWO-based deep residual network was 0.859, while the accuracies computed by LSS-CNN, RNN, SLKNN+MLKNN, SVM, NN, LSTM, and BPLion+LFNN were 0.824, 0.830, 0.839, 0.843, 0.849, 0.852, and 0.856, respectively.

Figure 5 Assessment of different techniques comparing with the proposed method by considering 20 Newsgroup dataset with mapper = 3. (A) TPR. (B) TNR. (C) Accuracy.

(b) Assessment with mapper = 4

Figure 6 presents the assessment of techniques measuring accuracy, TPR, and TNR and considering the 20 Newsgroups dataset with mapper = 4. The assessment of techniques with TPR measure is depicted in Fig. 6A. For chunk size = 3, the TPR evaluated by LSS-CNN, RNN, SLKNN+MLKNN, SVM, NN, LSTM, BPLion+LFNN, and the proposed RIWO-based deep residual network were 0.721, 0.769, 0.798, 0.801, 0.803, 0.806, 0.810, and 0.845, respectively. Likewise, for chunk size = 6, the TPR evaluated using LSS-CNN, RNN, SLKNN+MLKNN, SVM, NN, LSTM, BPLion+LFNN, and the proposed RIWO-based deep residual network were 0.810, 0.827, 0.836, 0.849, 0.840, 0.843, 0.846, and 0.859, respectively. The assessment of techniques measuring TNR is depicted in Fig. 6B. For chunk size = 3, the TNR evaluated by LSS-CNN, RNN, SLKNN+MLKNN, SVM, NN, LSTM, BPLion+LFNN, and the proposed RIWO-based deep residual networks were 0.831, 0.831, 0.842, 0.867, 0.848, 0.854, 0.859, and 0.870, respectively. Likewise, for chunk size = 6, the TNR evaluated by LSS-CNN, RNN, SLKNN+MLKNN, SVM, NN, LSTM, BPLion+LFNN, and the proposed RIWO-based deep residual network were 0.836, 0.839, 0.851, 0.857, 0.863, 0.866, 0.871, and 0.910. The assessment of the method measuring accuracy is depicted in Fig. 6C. For chunk size = 3, the accuracy evaluated using LSS-CNN, RNN, SLKNN+MLKNN, SVM, NN, LSTM, BPLion+LFNN, and the proposed RIWO-based deep residual network were 0.821, 0.831, 0.842, 0.846, 0.849, 0.854, 0.860, and 0.861, respectively. Likewise, for chunk size = 6, the accuracy evaluated using LSS-CNN, RNN, SLKNN+MLKNN, SVM, NN, LSTM, BPLion+LFNN, and the proposed RIWO-based deep residual network were 0.824, 0.838, 0.849, 0.852, 0.858, 0.863, 0.868, and 0.870, respectively. The performance improvements of LSS-CNN, RNN, SLKNN+MLKNN, SVM, NN, LSTM, BPLion+LFNN with respect to the proposed RIWO-based deep residual network and considering accuracy were 5.287%, 3.678%, 2.413%, 2.068%, 1.379%, 0.804%, and 0.229%, respectively.

Figure 6 Assessment of different techniques comparing with the proposed method by considering 20 Newsgroup dataset with mapper = 4. (A) TPR. (B) TNR. (C) Accuracy.

Comparative discussion

Table 3 shows the assessment of techniques in terms of accuracy, TPR, and TNR in the Reuters and 20 Newsgroups datasets. The Reuters dataset with mapper = 3 had a highest accuracy of 0.830 using the developed RIWO-based deep residual network, while the accuracies of the existing LSS-CNN, RNN, SLKNN+MLKNN, SVM, NN, LSTM, and BPLion+LFNN were 0.800, 0.812, 0.818, 0.818, 0.818, 0.819, and 0.819, respectively. The proposed RIWO-based deep residual network measured the maximum TPR of 0.925, while LSS-CNN, RNN, SLKNN+MLKNN, SVM, NN, LSTM, and BPLion+LFNN computed TPR of 0.846, 0.850, 0.865, 0.869, 0.878, 0.885, and 0.896, respectively. The proposed RIWO-based deep residual network computed the highest TNR of 0.880, while LSS-CNN, RNN, SLKNN+MLKNN, and BPLion+LFNN computed TNR of 0.824, 0.839, 0.849, 0.852, 0.856, 0.859, and 0.863, respectively. With mapper = 4, the highest TPR of 0.850, TNR of 0.940, and accuracy of 0.887 were found using the developed RIWO-based deep residual network. With the 20 Newsgroups datasets and mapper = 3, the highest TPR of 0.840, the highest TNR of 0.879, and the highest accuracy of 0.859 were computed by the proposed RIWO-based deep residual network. With mapper = 4, the highest TPR of 0.859, highest TNR of 0.910, and highest accuracy of 0.870 were determined using the developed RIWO-based deep residual network.

Table 3 Comparative discussion.

Datasets	Mappers	Metrics	LSS-CNN	RNN	SLKNN + MLKNN	SVM	NN	LSTM	BPLion + LFNN	Proposed RIWO-based deep residual network	
Reuters dataset	Mapper = 3	TPR	0.800	0.812	0.818	0.818	0.818	0.819	0.819	0.830	
		TNR	0.846	0.850	0.865	0.869	0.878	0.885	0.896	0.925	
	Accuracy	0.824	0.839	0.849	0.852	0.856	0.859	0.863	0.880	
	Mapper = 4	TPR	0.810	0.820	0.824	0.824	0.825	0.826	0.826	0.850	
		TNR	0.855	0.856	0.876	0.878	0.885	0.893	0.900	0.940	
Accuracy	0.833	0.849	0.852	0.857	0.859	0.863	0.868	0.887	
20 Newsgroup dataset	Mapper = 3	TPR	0.796	0.815	0.818	0.822	0.825	0.828	0.829	0.840	
		TNR	0.833	0.840	0.849	0.851	0.855	0.852	0.854	0.879	
	Accuracy	0.824	0.830	0.839	0.843	0.849	0.852	0.856	0.859	
	Mapper = 4	TPR	0.810	0.827	0.836	0.840	0.843	0.846	0.849	0.859	
		TNR	0.836	0.839	0.851	0.857	0.863	0.866	0.871	0.910	
Accuracy	0.824	0.838	0.849	0.852	0.858	0.863	0.868	0.870	

Conclusion

This article presents a technique for text classification of big data considering the MapReduce model. Its purpose is to provide a hybrid, optimization-driven, deep learning model for text classification. Here, pre-processing was carried out using stemming and stop word removal. The mining of significant features was also performed wherein SentiWordNet, contextual, and thematic features were mined from input pre-processed data. Furthermore, the selection of the best features was carried out using the Tanimoto similarity. The Tanimoto similarity examined the similarities between the features and selected the pertinent features with higher feature selection accuracy. Then, a deep residual network was employed for dynamic text classification. The Adam algorithm trained the deep residual network and dynamic learning was carried out with the proposed RIWO-based deep residual network and fuzzy theory for incremental text classification. Deep residual network training was performed using the proposed RIWO. The proposed RIWO algorithm is the integration of IWO and ROA, and it outperformed other techniques with the highest TPR of 85%, TNR of 94%, and accuracy of 88.7%. The proposed method’s performance will be evaluated using different datasets in the future. Additionally, bias mitigation strategies that do not directly depend on a set of identity terms and methods that are less dependent on individual words will be considered to effectively deal with biases tied to words used across many different contexts, e.g., white vs. black.

Supplemental Information

Supplemental Information 1 Python code

Click here for additional data file.

Supplemental Information 2 Dataset

Click here for additional data file.

Additional Information and Declarations

Competing Interests

Author Contributions

Data Availability

The authors declare there are no competing interests.

Hemn Barzan Abdalla conceived and designed the experiments, performed the experiments, analyzed the data, performed the computation work, prepared figures and/or tables, authored or reviewed drafts of the paper, and approved the final draft.

Awder M. Ahmed performed the experiments, analyzed the data, prepared figures and/or tables, and approved the final draft.

Subhi R.M. Zeebaree and Baha Ihnaini performed the computation work, prepared figures and/or tables, and approved the final draft.

Ahmed Alkhayyat performed the experiments, prepared figures and/or tables, and approved the final draft.

The following information was supplied regarding data availability:

The Python Code and the raw data are available in the Supplemental Files.

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
