# Peer review of "Rider weed deep residual network-based incremental model for text classification using multidimensional features and MapReduce"

_PeerJ Computer Science, doi:10.7717/peerj-cs.937_

## Round 0.1 · original submission · Major Revisions

From the comments of the reviewers, I think this is a valuable contribution. Please revise the paper accordingly, and then it will be evaluated again by the reviewers.

Additionally, Reviewer 1 has requested that you cite specific references. You may add them if you believe they are especially relevant. However, I do not expect you to include these citations, and if you do not include them, this will not influence my decision.

Reviewer 1 ·

Basic reporting

In this paper, author presented a MapReduce model for text classification in big data. However, there are some limitations that must be addressed as follows.
1. The abstract is very lengthy and not attractive. Some sentences in abstract should be summarized to make it more attractive for readers.
2. In Introduction section, it is difficult to understand the novelty of the presented research work. This section should be modified carefully. In addition, the main contribution should be presented in the form of bullets.
3. The most recent work about text classification and big data should be discussed as follows (‘An intelligent healthcare monitoring framework using wearable sensors and social networking data’, ‘Traffic accident detection and condition analysis based on social networking data’, ‘Fuzzy Ontology and LSTM-Based Text Mining: A Transportation Network Monitoring System for Assisting Travel’, and ‘Merged Ontology and SVM-Based Information Extraction and Recommendation System for Social Robots’).
4. It is better to merge subsection 2.1 and 2.2.
5. The authors should avoid the use of too many colors in figure (see figure1).
6. Equations should be discussed deeply.
7. Captions of the Figures not self-explanatory. The caption of figures should be self-explanatory, and clearly explaining the figure. Extend the description of the mentioned figures to make them self-explanatory.
8. The whole manuscript should be thoroughly revised in order to improve its English.
9. More details should be included in future work.

Experimental design

no comment

Validity of the findings

no comment

·

Basic reporting

The increasing demand for information and rapid growth of big data have dramatically increased textual data. The amount of different kinds of data has led to the overloading of information. For obtaining useful text information, the classification of texts is considered an imperative task. This paper develops a technique for text classification in big data using the MapReduce model. The goal is to design a hybrid optimization algorithm for classifying the text. This work is meaningful and potential in this field.

Experimental design

This paper develops a technique for text classification in big data using the MapReduce model.

Validity of the findings

The pre-pressing is done with the steaming process and stop word removal. In addition, the Extraction of imperative features is performed wherein SentiWordNet features, contextual features, and thematic features are generated. Furthermore, the selection of optimal features is performed using Tanimoto similarity.

Additional comments

1 This work should be polished by native English speaker. Some spelling and grammar mistakes should be avoided in this manuscript.
2 There are several typical machine learning classification model, such as SVM, neural network and so on. So, authors should compare the proposed method with other typical machine learning methods.
3 Some deep learning methods, including LSTM, should be compared with this method.
4 There are some typographical errors. Authors should polishted them.

---

## Round 0.2 · Major Revisions

Please improve your work accordingly.

·

Basic reporting

The increasing demand for information and rapid growth of big data has dramatically increased textual data. For obtaining useful text information, the classification of texts is considered an imperative task.

Experimental design

This paper develops a hybrid optimization algorithm for classifying the text. Here, the pre-pressing is done by the stemming process and stop word removal. In addition, the extraction of imperative features is performed, and the selection of optimal features is performed using Tanimoto similarity, which estimates
the similarity between the features and selects the relevant features with higher feature selection accuracy. After that, a deep residual network trained by the Adam algorithm is utilized for dynamic text classification. In addition, the dynamic learning is performed by the proposed Rider invasive weed optimization (RIWO)-based deep residual network along with fuzzy theory. The proposed RIWO algorithm combines Invasive weed optimization (IWO) and the Rider optimization algorithm (ROA).

Validity of the findings

The aim is to devise an optimization-driven deep learning technique for classifying the texts using the MapReduce framework. Initially, the text data undergoes pre-processing for removing unnecessary words. Here, the pre-processing is performed using the stop word removal and stemming process. After that, the features, such as SentiWordNet features, thematic features, and contextual features, are extracted. These features are employed in a deep residual network for classifying the texts. Here, the deep residual
network training is performed by the Adams algorithm. Finally, dynamic learning is carried out wherein the proposed RIWO-based deep residual network is employed for incremental text classification. Here, the fuzzy theory is employed for weight bounding to deal with the incremental data.

Additional comments

This paper develops a hybrid optimization algorithm for classifying the text. Here, the pre-pressing is done by the stemming process and stop word removal. In addition, the extraction of imperative features is performed, and the selection of optimal features is performed using Tanimoto similarity, which estimates
the similarity between the features and selects the relevant features with higher feature selection accuracy. This work is meaningful and potential in this field.
1 Some errors, including spelling errors and grammar ones should be polished.
2 Some typical references should be discussed in this work.

---

## Round 0.3 · accepted · Accept

I read the paper and the response letter. I believe the paper has been revised well.